# Detection and quantification of key dental pathogens through wastewater monitoring

Olivia N. Birch, Sang C. Par, Justin C. Greaves*

Department of Environmental and Occupational Health, School of Public Health Indiana University-Bloomington

* jcgreave@iu.edu

## Abstract

Wastewater-based epidemiology (WBE) has been widely used to track viral pathogens like SARS-CoV-2 and polio, but its potential for monitoring common dental bacterial pathogens that infect the oral cavity has yet to be explored. *Streptococcus mutans* and *Porphyromonas gingivalis* are key oral bacterial pathogens that cause highly prevalent dental diseases worldwide, such as dental caries and gingivitis. Our main objective for this study was to investigate the presence and prevalence of these oral bacteria in wastewater to determine the feasibility of using WBE for oral pathogens. We measured *S. mutans* and *P. gingivalis* nucleic acids in weekly samples for 24 months at a local wastewater treatment plant. Between June 2023 and May 2025, a weekly sample of untreated wastewater was collected, resulting in a total of 100 samples collected over the timespan. Samples were concentrated, extracted for DNA, and then tested for each bacterium. Our results showed that 89% and 58% were positive for *S. mutans* and *P. gingivalis*, respectively, which shows that wastewater surveillance is appropriate for oral bacteria. Average concentrations were 4.57 $\log_{10}$ genome copies/L and 3.03 $\log_{10}$ genome copies/L for *S. mutans* and *P. gingivalis*, respectively. Detections of oral bacteria were observed in the primary and final effluent, but concentrations were significantly lower in the final effluent than in the untreated wastewater. The high levels of oral bacteria in wastewater indicated a potential transmission mechanism for these bacteria through water, specifically for *S. mutans*. Additionally, this study underscores the unique potential for WBE to be used in the surveillance of oral bacterial pathogens.

## 1. Introduction

Dental caries, or tooth decay, is one of the most prevalent chronic diseases worldwide, affecting nearly 2.3 billion people with permanent tooth decay and over 530 million children with primary tooth loss, according to the World Health Organization [1]. *S. mutans* is a major culprit in dental caries, playing a significant role in plaque

**Data availability statement:** All relevant data are within the manuscript and its Supporting Information files.

**Funding:** The author(s) received no specific funding for this work.

**Competing interests:** The authors have declared that no competing interests exist.

formation and acid production that demineralizes tooth enamel [2–4]. *P. gingivalis*, on the other hand, is a key pathogen in periodontal disease, a severe gum infection that can lead to tooth loss and systemic health complications, which can include cardiovascular disease and diabetes [5,6]. The high global burden of these oral health issues underscores the importance of developing innovative disease surveillance methods to understand the spread of these diseases.

Wastewater-based epidemiology (WBE) has become increasingly popular in recent years as a tool to monitor the spread of endemic and emerging pathogens passively in a community. WBE has already been successfully employed to monitor various pathogens, such as SARS-CoV-2 and antimicrobial resistance genes, yet its application to oral bacteria remains largely unexplored [7,8]. Given the widespread presence of these oral bacteria in human populations, understanding their prevalence in wastewater can provide insights into community-level oral health trends and transmission dynamics. Traditional surveillance methods, such as clinical examinations and self-reported surveys, have limitations in reach, accuracy, and cost-effectiveness [9,10]. WBE, on the other hand, allows for large-scale, community-wide monitoring without the need for individual participation, offering a more comprehensive and real-time picture of oral health burdens [11,12]. It has also been determined in previous studies that wastewater includes pathogens shed by sputum as well, which could be a strong access route for these dental pathogens [13]. If a strong correlation between wastewater bacterial levels and oral disease prevalence can be established, then this approach could serve as an early warning system for emerging trends in oral health conditions, prompting timely public health interventions [14,15]. Current approaches to monitoring oral health at the population level rely heavily on dental claims data and self-reported surveys. However, these methods have notable limitations. Claims data often exclude individuals without access to dental insurance, while self-reported surveys are subject to recall and reporting biases and are typically conducted infrequently [16]. As a result, these approaches may fail to capture timely, representative data, especially among vulnerable populations who already face barriers to dental care and experience higher burdens of oral disease. However, there is no current public health surveillance system for the oral diseases caused by these specific dominant pathogens at the national or local level. Hence, our current understanding of the prevalence of these diseases within the past 5 years has been severely limited. Therefore, the use of WBE is especially timely in filling the gaps in information about these oral pathogens.

Beyond monitoring, the detection of oral bacteria in wastewater raises concerns about their environmental persistence and potential transmission pathways [17,18]. Although wastewater treatment processes are designed to remove or reduce microbial contaminants, some pathogens can survive or be reintroduced into the environment through treated effluent and biosolids by attaching to particles [6,19]. If *S. mutans* and *P. gingivalis* demonstrate resilience through treatment processes, there is a risk of environmental exposure, which could have implications for public health [20–24]. Further studies are needed to assess whether these bacteria can colonize environmental surfaces, influence microbial ecosystems, or contribute to antimicrobial resistance.

In this study, we determined the concentration of *S. mutans* and *P. gingivalis* in wastewater samples over one year. We also determined the concentration of *S. mutans* and *P. gingivalis* in wastewater primary influent, primary effluent, and secondary effluent. By measuring the prevalence of *S. mutans* and *P. gingivalis* in untreated wastewater and evaluating their reduction through treatment processes, this study aims to bridge a significant knowledge gap in WBE. If successful, integrating oral bacteria surveillance into WBE frameworks could revolutionize how we monitor, predict, and mitigate oral health diseases at a population level. These insights could be used to inform or evaluate public health interventions such as community water fluoridation programs, school-based dental screenings, or targeted outreach in high-incidence areas [25]. Importantly, integrating WBE into oral health surveillance frameworks could help address long-standing inequities by identifying underserved populations that might otherwise be missed using conventional data sources. This capability supports a more proactive and inclusive approach to public oral health planning and resource allocation. Wastewater-based epidemiology (WBE) represents a complementary and potentially more equitable surveillance tool. Because WBE captures pooled biological signals from entire communities regardless of socioeconomic or insurance status, it has the capacity to provide anonymized, near real-time insights into population-level trends in oral pathogen circulation.

## 2. Materials and methods

### Study site and temporal sampling

100 sewage samples were collected from Blucher Poole Wastewater Treatment Plant, a local wastewater treatment plant (WWTP) in Bloomington, Indiana. Blucher Poole has a flow of more than 4.5 million gallons per day and caters to the northern part of Monroe County with an estimated total catchment population of 25,000 individuals. This treatment plant was selected to represent the majority of the Bloomington, Indiana, population and to observe bacterial shedding rates in wastewater. These samples were collected weekly from June 2023 to May 2025. A second local WWTP, the Dillman Road Wastewater Treatment Plant, was selected to enable comparison and contrast between the bacterial quantification of the two plants. Dillman Road is located on the south side of Bloomington, Indiana, and has a peak hydraulic capacity at 30 million gallons daily with an estimated total catchment population of 55,000 individuals [26]. For this WWTP, 13 weekly sewage samples from December 2023 to February 2024 were collected and transported to the lab. 24-hour composite influent (prior to primary treatment) wastewater samples were collected in 500-milliliter sterile polypropylene bottles and transported in a cooler and frozen to −20°C within 2 hours of collection. Once received, these samples are immediately prepared for filtration.

### Effluent sampling

Composite samples of primary and secondary/final effluent wastewater were collected from the local WWTP and placed at −20°C for storage until sample processing. These samples were collected during a 2-week intensive study and in the month of February when concentrations were highest. For the primary influent, a 500 mL 24-hour composite sample was collected; this is the sewage sample that comes directly into the treatment plant and has not received any treatment. We received a 100 mL sample of the 24-hour composite primary effluent sample, which consists of the sewage sample after primary sedimentation has occurred. For the 24-hour composite secondary/final effluent, 500 mL was collected in 500 mL sterile polypropylene bottles. This sample is the sewage sample post aeration, activated sludge, and secondary sedimentation. All samples were transported in the same manner as mentioned previously.

### Concentration and DNA extraction

For final effluent samples, 250 mL of the final effluent was taken to be filtered, and for all other samples, 50 mL was prepared to be filtered. All samples were acidified to a pH of 3.5 by the addition of 20 μL of Hydrochloric Acid to increase the electrostatic attraction of the bacteria inside the sample to the negatively charged filters. A known concentration of

bovine coronavirus solution was also added to each sample as a tracer. For filtration, an electronegative mixed cellulose ester (MCE) 47 mm diameter and 0.45 μm pore size filter was used to filter samples based on previous studies [8,27,28]. Once filtration through the filter was complete, the filters were then transferred into 2.0 mL PowerBead tubes and either further extracted immediately or stored at −20 °C for extraction later. The DNA from the samples was then extracted in accordance to the provided manufacturer's instructions using the AllPrep PowerViral DNA/RNA extraction kit (Cat. No. 28000−50 Qiagen, Germantown, MD,USA). This kit resulted in the elution of DNA in 100 μL of RNase-free water provided within the kit. Once eluted, these samples were then stored at −20°C in preparation for quantification.

## Bacterial and viral quantification

Bacterial quantification of the samples was performed using digital polymerase chain reaction (dPCR) and the QIAcuity™ Four Platform System (Cat. No 911042, Qiagen, Germantown, MD, USA), the QIAcuity Nanoplate 26k (Cat. No 250001, Germantown, MD, USA), and the QIAcuity™ Probe PCR Kit (Cat. No. 250132 Qiagen, Germantown, MD, USA) as described in previous studies [8]. The environmental mix for each molecular testing consisted of 1X Qiacuity Probe PCR mix, 08 μM of each forward and reverse primer, 0.4 μM of each probe, and 10 μL of the target DNA for a total volume of 40 μL. The molecular primers and probes used for the detection of *S. mutans* and *P. gingivalis* are described in S1 Table. While the primer and probe sets were chosen from peer-reviewed, previously validated assays, their specificity was formally assessed through in silico screening using NCBI BLAST. Primer and probe sets were also able to detect a diluted culture of each bacterium. All samples were tested using a duplex, where *S. mutans* was detected in the green channel with the FAM probe and *P. gingivalis* was tested in the yellow channel with the HEX probe. The primers and probes were manufactured by IDT (Coralville, IA, USA). The Zen™ internal quencher (IDTDNA.com) and the 3' Black Hole ™ quencher were used to double-quench the probes. PCR was run using the following conditions: an initial denaturation step at 95 °C for 2 minutes, then 45 cycles of 95 °C for 5 seconds and 60 °C for 30 seconds. The lowest detection limit was estimated for each PCR assay using the value of one positive well partition from DNA controls, which resulted in a calculated 380 copies/L for both *S. mutans* and *P. gingivalis* [7]. All non-detection samples were reported as the limit of detection. CrAssphage was used as a molecular fecal indicator and quantified with techniques, probes, and primers published previously [7,29,30]. The threshold for each PCR reaction was marked by the top of the negative band of the negative control for the specific probe being tested.

## Data analysis and statistics

The following formula was used to calculate the concentration of oral bacteria in gene copes/L:

$$\frac{gc}{L} = \frac{gc}{uL} X \frac{Vol\ PCR\ reaction\ (40uL)}{Vol.\ NA\ analyzed\ (5uL)} X \frac{Vol.\ NA\ eluted\ (50uL)}{vol\ sample\ (0.05L)} \times correction\ factor$$

The flow rate and population (population normalization done through crAssphage) correction factor were calculated by determining the average flow rate or crassphage concentrations across the year, then dividing the monthly values by the annual average value. Each sample concentration was then multiplied by both the flow rate and population correction factors. The analyses for our data were conducted using ANOVA and were performed on GaphPad Prism v 10 (Boston, MA, USA) where independent factors were taken into consideration. Graphs and figures were made using GraphPad Prism v 10 (Boston, MA, USA).

## Sequencing of dental pathogens

Amplicon sequencing was done for the detection of *S. mutans* and *P. gingivalis* using primers developed in this study. For *S. mutans,* the targeted gene was glucosyltransferase-I (gtfB) using the forward and reverse primers ATCATTACGTCTGTCCGCTATG and CATCGGCTGTCCCGTATTT, respectively. For *p. gingivalis,* the targeted gene was 16SrRNA and the

forward and reverse primers used were CCCGTTGAAAGACGGACTAAA and CTTCAGTGTCAGTCGCAGTATG, respectively. The amplification for *S. mutans* resulted in a 691-nt product. The amplification of *P. gingivalis* produced a 606-nt product. For the qPCR, each reaction contained 10 µL of HotStarTaq® Plus Master Mix (Qiagen, Cat. No. 203645; Hilden, Germany), 2 µL of 1×HST buffer, and 5 µL of template DNA. The thermal cycling program began with a 15-minute activation at 95 °C, followed by 50 PCR cycles with denaturation at 95 °C for 5 seconds, annealing at 55 °C for 30 seconds, and extension at 72 °C for 1 minute.

To confirm amplification specificity of the PCR products, samples were checked by a run through agarose gel electrophoresis and positive controls (gblock of the gene of target). Products that produced the expected bands were purified and subjected to Sanger sequencing by Eurofins Genomics. The obtained sequences were compared to the NCBI nucleotide database using BLAST, and the top seven matches for each sample were retrieved for subsequent analysis. Multiple sequence alignment was performed with the ClustalW tool in NCBI Genome Workbench, and phylogenetic relationships were inferred using the maximum-likelihood method implemented in MEGA XI.

## 3. Results

### Bacterial and viral detection and quantification

Fig 1 shows the average monthly wastewater flow rates into the main local wastewater treatment plant tested in this study. Values throughout the year show the lowest value being in the summer and the highest values in the winter months. In addition to monitoring flow rates, the fecal indicators crAssphage was measured in all the samples throughout the duration of the experiments and concentrations remained stable over the two year period (S1 Fig). CrAssphage had an average detected concentration of 8.50 $\log_{10}$ GC/L. Statistical tests were done using a one-way ANOVA followed by tukey multiple comparisons to measure changes in monthly concentrations crAssphage, and results showed no statistical differences between crAssphage concentrations over the different months of the experiment.

Fig 2 shows the monthly average concentrations over the span of two years for *S. mutans* and *P. gingivalis* in the main local wastewater treatment plant. Bacterial concentrations were normalized by wastewater flow rates to accommodate for population changes. The highest concentration of *S. mutans* was observed in February of year 1, with a calculated monthly average concentration of 5.01 $\log_{10}$ GC/L, and the lowest detected concentration was in August of year 2, with 3.39 $\log_{10}$ GC/L. For *P. gingivalis*, the month with the highest average concentration was November 2023, with a calculated average monthly concentration of 3.81 $\log_{10}$ GC/L, and the month with the lowest values was September 2024, where none of the samples had detections and the values were assumed to be the detection limit. Over the course of the

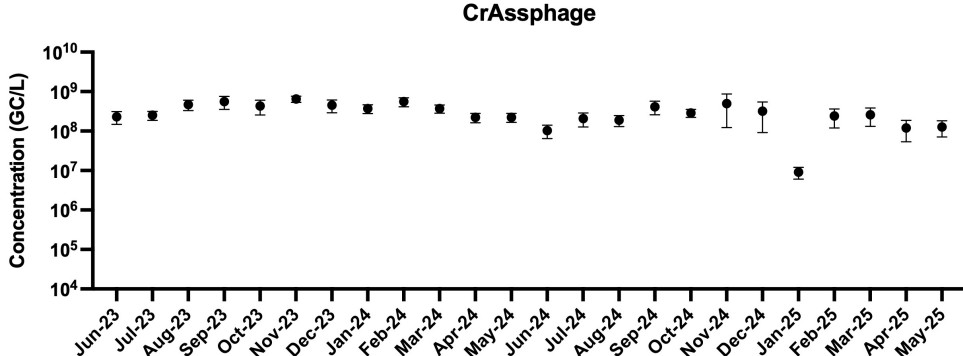

**Fig 1. Monthly average concentrations of crAssphage in Bloomington wastewater over the two-year duration.** The bars are standard deviation for four samples collected each month (n = 4-5).

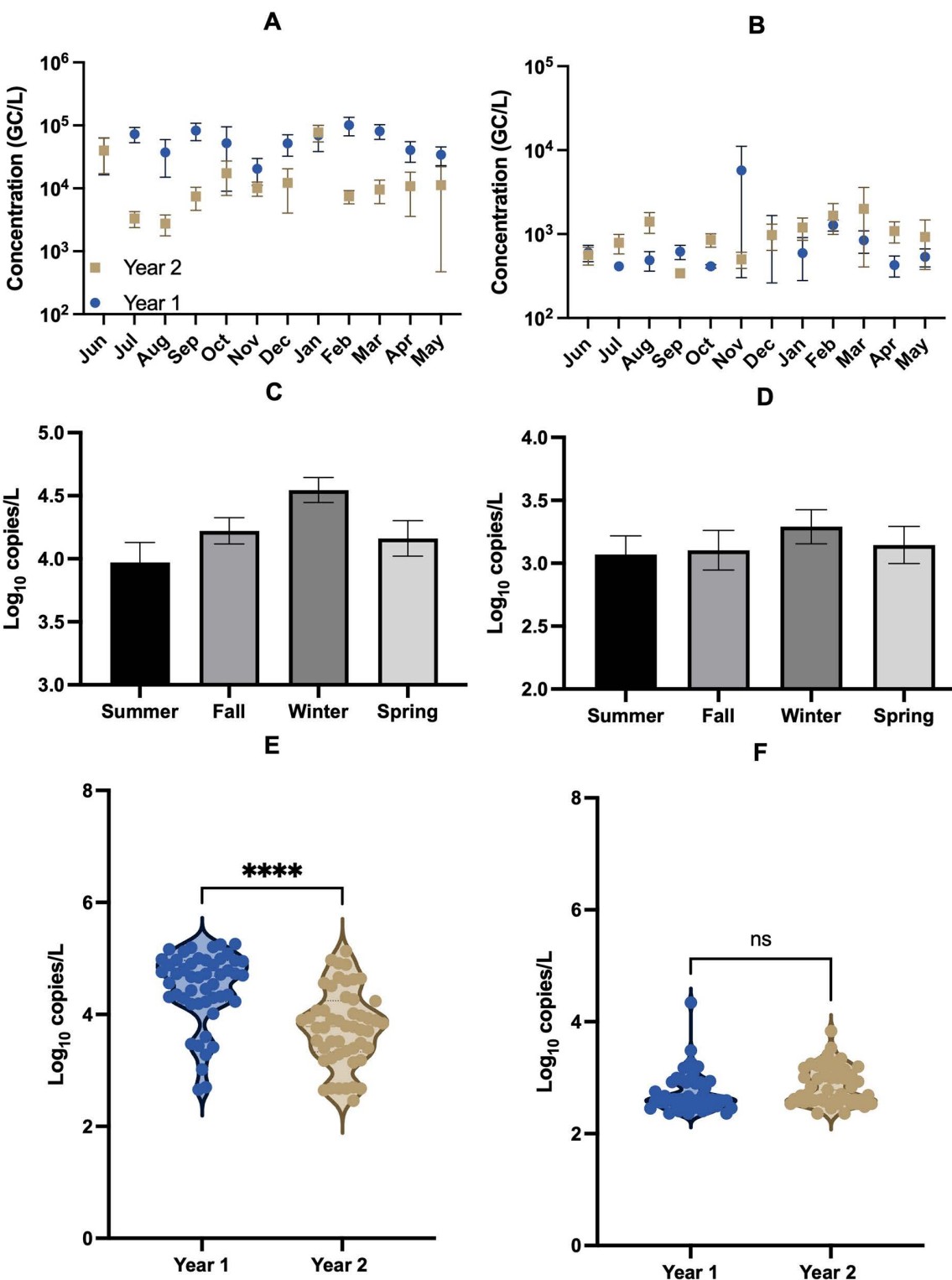

**Fig 2. Concentrations of oral bacteria in wastewater.** Top two figures show monthly average concentrations of *S.mutans* (A) and *P. gingivalis* (B) in Bloomington wastewater over the two-year duration normalized by the change in wastewater flow. The bars are standard deviation for four samples collected each month (n = 4-5). Middle figures show average concentrations for *S.mutans* (C) and *P. gingivalis* (D) in each season over the two years. Bottom figures show violin plots for both oral bacteria in year 1 and year 2 (E for *S.mutans* and F for *P. gingivalis*).

entire selected timeline, the average detected concentration of *S. mutans* and *P. gingivalis* was 4.57 $\log_{10}$ GC/L and 3.03 $\log_{10}$ GC/L, respectively. **Table 1** breaks down the concentration values for *S. mutans* and *P. gingivalis* into four different time periods within the year, corresponding with Spring (March-May), Summer (June-August), Fall (September-November), and Winter (December-February). Values are also the average of the two years. For *S. mutans,* the highest average detected concentration was in the winter with a value of 4.78 $\log_{10}$ GC/L and a positive detection rate of 100%. Statistical tests showed winter concentrations were significantly higher than summer and spring concentrations for *S. mutans*. *P. gingivalis* also had the highest average detected concentration in the winter, with a value of 3.12 $\log_{10}$ GC/L and a positive detection rate of 84% which was also the highest detection rate across all seasons. No significant differences were found between the concentrations across all seasons for *P. gingivalis*.

A comparison was also done between year 1 (June 2023 to May 2024) and year 2 (June 2024 to May 2025) using Pearson's correlation r. Statistical analysis showed no correlation between the different years for both bacteria (S3 Fig). Correlation analysis for crAssphage for the different years showed a strong positive correlation (r = 0.77). A comparison was made between two different WWTPs, Blucher Poole (BP) and Dillman Road (DR), for the two bacteria in the winter of year 2, which can be observed in S2 Fig. For both WWTPs, *S. mutans* was observed to have a larger spread in concentrations than *P. gingivalis,* which can be observed in the top two graphs within S2 Fig. More samples of *P. gingivalis* were disregarded due to not being detected, in comparison to *S. mutans*. For BP, the average concentration of *S. mutans* was 4.93 $\log_{10}$ GC/L, and for *P. gingivalis,* it was 3.14 $\log_{10}$ GC/L. The average concentration of *P. gingivalis* for DR was higher at 3.53 $\log_{10}$ GC/L, whereas *S. mutans* was relatively similar to BP with an average concentration of 4.86 $\log_{10}$ GC/L. *P. gingivalis* was statistically higher in the DR WWTP compared to the BP WWTP, whereas there were no statistical differences between the concentrations of *S. mutans* in the DR and BP WWTPs.

## Effluent sampling

In **Fig 3**, the concentrations of *S. mutans* and *P. gingivalis* in primary influent, primary effluent, and secondary effluent are shown. For both species of oral bacteria, they were most prevalent in the primary influent samples, with *S. mutans* having an average concentration of 3.87 $\log_{10}$ GC/L and for *P. gingivalis*, a concentration of 3.21 $\log_{10}$ GC/L. The difference in concentration between the primary influent and the primary effluent samples was larger for *P. gingivalis* than it was for *S. mutans,* with calculated differences of 1.08 $\log_{10}$ GC/L and 0.106 $\log_{10}$ GC/L, respectively. The final effluent samples still resulted in some detected concentrations, with *S. mutans* having an average of 3.08 $\log_{10}$ GC/L and *P. gingivalis* having an average of 1.99 $\log_{10}$ GC/L. The difference between primary influent and secondary effluent for both *S. mutans* and *P. gingivalis* was significant (p < 0.0001). For *S. mutans*, the average concentration for primary effluent and secondary effluent was determined to be significant as well (p < 0.0001). The difference between the concentration of *P. gingivalis* in

**Table 1. Detection rates and average concentrations of *S. mutans* and *P. gingivalis* in wastewater samples.**

| Target | Seasons | Sample period | No. of positive samples (%) | Concentration (Log₁₀ copies/L) |
|---|---|---|---|---|
| S. mutans | Spring 2024 & 2025 | March-May | 19 (76) | 4.37 |
| | Summer 2023 & 2024 | June-August | 20 (80) | 4.34 |
| | Fall 2023 & 2024 | Sept-Oct | 25 (100) | 4.35 |
| | Winter 2024 & 2025 | Jan-Feb & Dec previous year | 25 (100) | 4.78 |
| | Total | | 89 (89) | 4.57 |
| P. gingivalis | Spring 2024 & 2025 | March-May | 12 (48) | 2.93 |
| | Summer 2023 & 2024 | June-August | 12 (48) | 2.75 |
| | Fall 2023 & 2024 | Sept-Oct | 13 (52) | 3.04 |
| | Winter 2024 & 2025 | Jan-Feb & Dec previous year | 21 (84) | 3.12 |
| | Total | | 58 (58) | 3.03 |

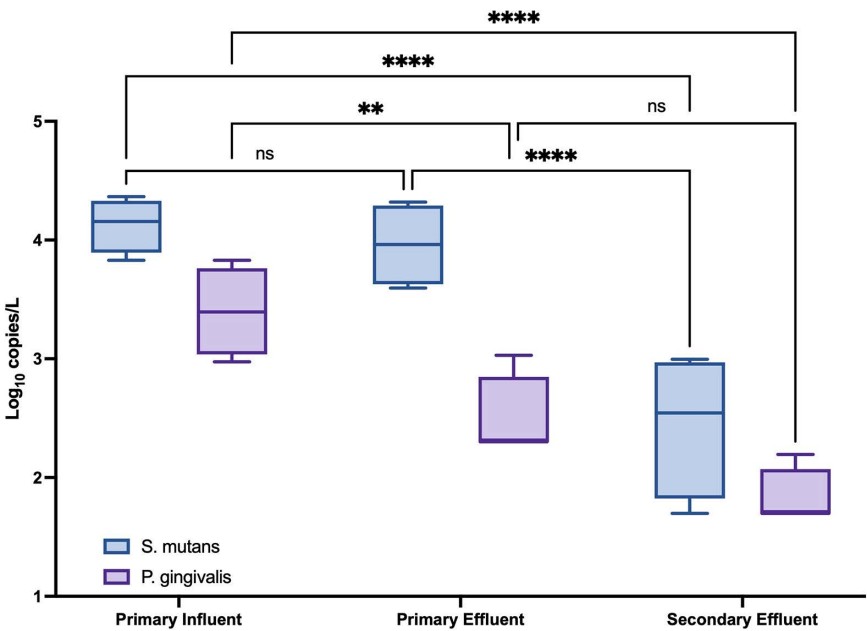

**Fig 3. Occurrence of *S. mutans* and *P. gingivalis* in wastewater treatment plant primary influent, primary effluent, and secondary effluent in 4 replicate samples.**

primary influent and primary effluent samples was significant (p = 0.009). There was determined to be no significance in the concentration of *S. mutans* in the primary influent and the primary effluent samples (p = 0.816), as well as there being no significant difference in the concentration of *P. gingivalis* between the primary effluent and secondary effluent samples (p = 0.064).

## Dental pathogen sequencing results

The use of the nested primer that targeted the gtfB gene for *S. mutans* resulted in 12 samples successfully amplified, which were then sequenced. Once sequenced, BLAST results of the sequences confirmed that all 12 were *S. mutans*. The nested primer strategy used for the 16SrRNA gene in *P. gingivalis* resulted in 1 sample successfully amplified. This PCR product was sequenced, and BLAST confirmed it to be *p. gingivalis*. The phylogenetic analyses of the gtfB gene of *S. mutans* and the 16SrRNA gene of *P. gingivalis* are shown in **Fig 4**.

For *S. mutans,* the analysis placed all Indiana wastewater samples sequenced into a single, well-supported monophyletic group. This cluster was most closely related to oral reference sequences from New York (2013), Iraq (2023), and China (2022), indicating that the gene sequenced from Indiana wastewater samples was part of a broader, globally circulating lineage. Within the Indiana group, samples collected across different months showed minimal genetic divergence, suggesting little change within the bacteria over the study period and that one strain dominates within this area. Other oral reference strains from Japan (1998–2024), England (2018), and U.S. isolates (Illinois 1988, Oklahoma 2002, Florida 2020) formed more distant groups, consistent with longer-term evolutionary separation. Phylogenetic analysis of the *P. gingivalis* 16SrRNA gene revealed that the Indiana wastewater isolate collected on June 2, 2025, clusters most closely with oral reference sequences from Ohio (2017) and Bern, Switzerland (2002). This subgroup joins a larger clade that includes additional North American sequences from Florida (2015) and New York (2006), as well as more distantly related Asian isolates from China (2013, 2023) and Iran (2016). The moderate bootstrap support (42–55) suggests recent shared ancestry with the Ohio groups while maintaining genetic distinction from older European and Asian lineages.

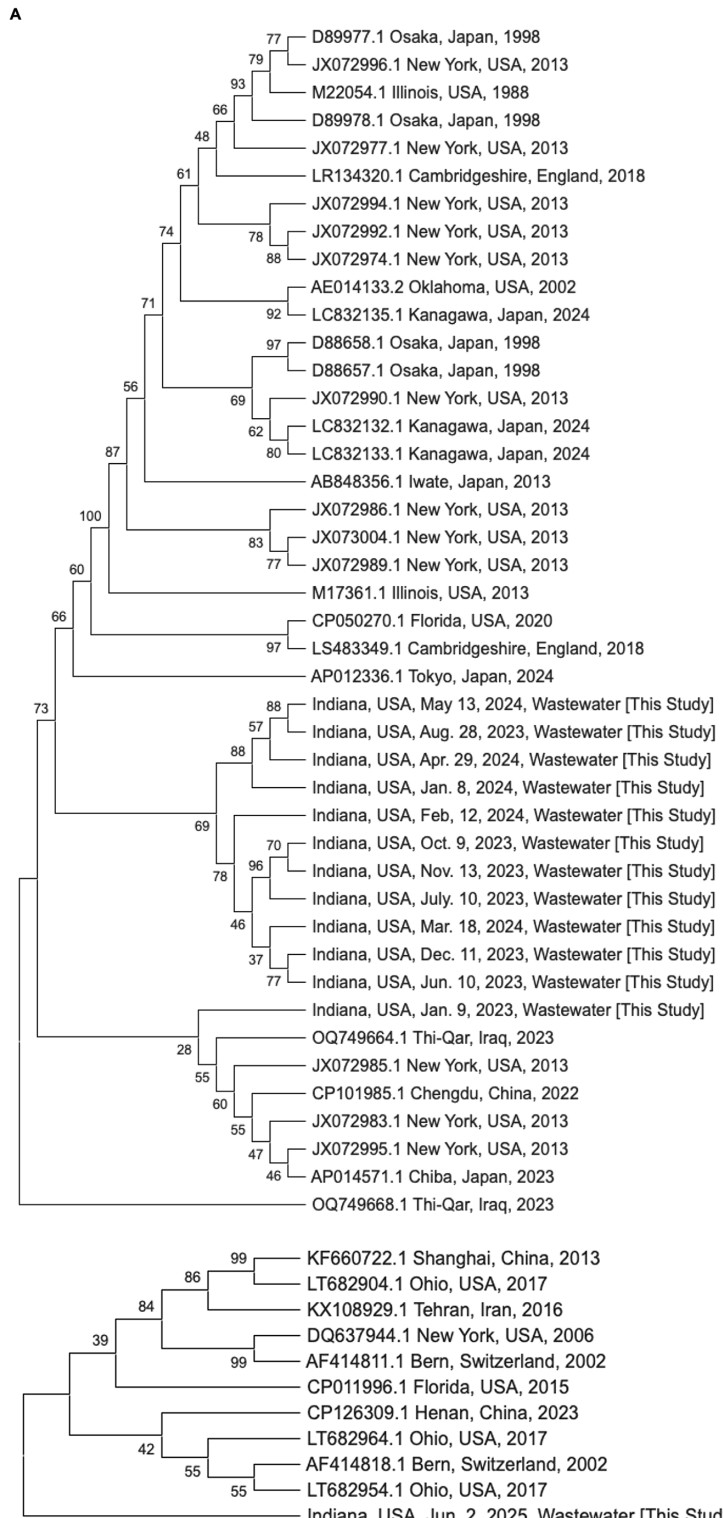

**Fig 4. Phylogenetic analysis in wastewater samples for *S. mutans* (A) and *P. gingivalis* (B).** Numbers represent bootstrap values from maximum likelihood analysis. Sequences in this study were compared with the top 10 sequences obtained from NCBI BLAST search of each wastewater sample.

## 4. Discussion

To the best of the authors' knowledge, this study is the first study to detect and quantify *S. mutans* and *P. gingivalis* in wastewater samples. These two pathogens cause two of the most common dental diseases that infect a majority of populations worldwide. Even though these pathogens are highly prevalent in US populations, routine surveillance on the prevalence of these different pathogens or the oral disease they cause is not done, especially since the outbreak of the COVID-19 pandemic [31,32]. Through our study, we were able to show that both pathogens were detected in more than 60% of the 65 samples tested in wastewater over the 4 different seasons of the year. The high detection of these oral pathogens in wastewater samples from a mid-size city (75,000 residents) shows that wastewater-based epidemiology (WBE) is highly feasible for these dental pathogens and oral health. WBE may also serve as a low-cost method to monitor the spread and prevalence of these dental diseases when traditional monitoring methods are not used.

*S. mutans* had consistently higher concentrations than *P. gingivalis* in wastewater throughout the sampling period. In some samples, concentrations of *S. mutans* reached more than 10 times higher than *P. gingivalis*. With numerous studies estimating higher percentages of tooth decay (90%) than gum disease (50%) in adults [33,34], our differential observations of *S. mutans* (which associates with tooth decay) and *P. gingivalis* (which associates with gum disease) is consistent with past clinical prevalences of these diseases in US populations [35–38]. Higher prevalences of a disease should result in higher concentrations in wastewater [39,40]. However, concentrations of *S. mutans* could also be higher in wastewater due to the shedding dynamics from teeth as opposed to the shedding dynamics of *P. gingivalis* from gums [41–43]. While studies have not specifically tested shedding of each pathogen, previous studies have observed statistically higher concentrations of *S. mutans* than *P. gingivalis* in dental samples from healthy patients, alluding to higher shedding of *S. mutans* than *P. gingivalis* [32,44,45]. Differential shedding could be due to the stark difference in oral hygiene when it comes to the gums and teeth, where cleaning of the teeth is done more frequently than cleaning of the gums [46–49]. The variations in shedding might also be simply attributed to the ease with which they attach or detach from different surfaces, such as enamel versus gums [50–52].

In addition to being shed in saliva, oral pathogens may also be swallowed, survive passage through the gastrointestinal tract, and be excreted in feces. Several studies have detected the DNA of oral microbes, including S. mutans, in human feces, particularly through 16S rRNA gene sequencing [53,54]. This fecal shedding pathway may significantly influence the concentration of these bacteria in wastewater. For example, *S. mutans* may be more resilient than *P. gingivalis* during gastrointestinal transit, potentially resulting in higher levels of S. mutans in fecal matter. While feces are generally considered the primary route of microbial contribution to wastewater, no dedicated studies have confirmed the specific shedding patterns of *S. mutans* and *P. gingivalis* [55]. As a result, there remains uncertainty about how their wastewater concentrations correlate with actual infection prevalence in the population. Future studies should examine the shedding dynamics of each oral pathogen so that improved connections between dental health and wastewater concentrations can be developed.

### Seasonal patterns

WBE has been shown to be a very useful tool for understanding disease trends, specifically over the different seasons of a year [56,57]. When trends for different seasons are determined, health agencies can then pick specific seasons throughout the year to focus mitigation efforts on [14,58]. Our results showed seasonal trends for *S. mutans* and *P. gingivalis,* with both oral bacteria having their highest average concentrations in the winters over the two-year period. However, results for *P. gingivalis* were not significant, potentially due to the low detection rate of this pathogen. Nevertheless, detections in winter for *P. gingivalis* were also much higher than the detections in other months, suggesting potential seasonality similar to *S. mutans*.

*S. mutans* showed significantly higher concentrations in the winter than in the spring and summer. The highest temperature months of July and August exhibited notably lower concentrations compared to the other months of the year, especially in year 2. These results highlight that *S. mutans* and dental caries could be more prevalent in months with

colder weather. Prior studies have observed diminished oral health in colder months due to drier mouths and immune system changes in the winter [59,60]. Drier mouths due to increased indoor heater use can reduce the volume of saliva (known to prevent bacterial growth), which in turn can cause the extended growth of pathogens like *S. mutans* [61–63]. Colder months can also lead to diet changes, lowering nutrient levels, or less hydration, which could potentially contribute to tooth decay and poorer oral health in colder months compared to hotter months [59,64,65]. Although we observed higher average concentrations of oral pathogens during the winter months across both years, several outlier months in each season deviated from this trend, suggesting that additional factors may be influencing oral health dynamics. These findings are also region-specific; other areas may exhibit different seasonal patterns for these two oral pathogens. Future studies should integrate oral health indicators with behavioral and environmental factors to uncover the underlying drivers of seasonal variation.

Though average concentrations in the different seasons across years show potential seasonal trends, trends of bacteria in each year were not well correlated (S3 Fig). Specifically for S. mutans, our data observed significantly lower concentrations in year 2 than in year 1, suggesting a reduction in oral pathogen burden in the community. P. gingivalis has statistically similar concentrations across both years, but concentrations over time were also not well correlated. A potential reason for the reduction in oral health burden is that COVID-19-related dental clinic closures and delayed care likely worsened oral health, but as services resumed, individual dental health may have improved, potentially lowering oral health-related markers in wastewater over time [66,67]. Prior clinical studies have also shown a lowering oral health burden over the past decade, which follows the trend in our study [68–70]. Additionally, year-to-year patterns can significantly change depending on healthcare access, economic conditions, and public health policies that affect community oral health in a year [71–73].

## Variation in WWTP

Our study results showed that *S. mutans* concentrations in wastewater from both WWTPs were not statistically different during the wintertime, even though both WWTPs serve different population numbers and types. These results suggest that *S. mutans* can be prevalent in different WWTP and populations across geographical areas. The higher observed concentration of *S. mutans* compared to *P. gingivalis* in the DR WWTP is consistent with what was observed for BP WWTP. Our results do show statistically higher concentrations of *P. gingivalis* in the DR WWTP compared to the BP WWTP (p = 0.0273). BP WWTP primarily serves the university campus community, which is mostly comprised of young students between the ages of 18 and 24. DR WWTP serves the rest of the local population, which includes children and vulnerable groups that may not be present in the other WWTP. As oral health significantly impacts those who are most vulnerable, such as children (below 18) and the elderly (above 65), it is expected that there would be higher levels of reduced oral health in the DR WWTP, which contains these vulnerable groups [26,62,74]. However, to specifically determine whether these factors affect oral health, further analysis is needed on samples across the sewer shed and in other geographic locations.

## Fate in the wastewater treatment process

Throughout the wastewater treatment process, both *S. mutans* and *P. gingivalis* were detectable but at significantly lower concentrations. These data suggest that the treatment process is reducing the oral bacterial population, but not completely. The primary treatment step was able to significantly reduce *P. gingivalis,* but the secondary treatment step (both activated sludge and secondary sedimentation process) was needed to reduce *S. mutans*. These differences are potentially due to how likely each microbe is to stick to fecal particles in wastewater. Multiple prior studies have shown a reduction of various pathogens in wastewater through the primary and secondary treatment processes [75–78]. The same studies also show that certain pathogens are able to survive through the process and enter the environment at lower levels, similar to what we are observing for *P. gingivalis* and *S. mutans* [75–79]. It has been proven in past studies that

the possibility of horizontal transmission of dental pathogens, like *S. mutans*, is possible, whether this be through kissing, shared drinks, or other forms of contact with infectious fluids [80,81]. Although the risk of transmission of dental pathogens from wastewater effluent to the oral cavity has not been fully studied, this risk can be assessed similarly to that of enteric viruses entering via the fecal oral route. When considering an acceptable or required reduction efficiency in the wastewater treatment process, future studies should examine the relationship between bacterial load and the risk of contracting an infection, as well as the infectivity of the target pathogens identified. This observation of wastewater treatment plant survival, therefore, highlights the potential novel transmission pathway of these oral pathogens through the environment in contaminated waters.

### Dental pathogen sequencing in wastewater

The phylogenetic analysis of *S. mutans* sequences obtained from Indiana wastewater revealed that all study samples grouped within established monophyletic clades of *S. mutans*. Most sequences clustered closely with isolates previously reported from New York, Iraq, and China, while others aligned with earlier U.S. and Japanese strains. Similar patterns of widespread circulation have been documented in clinical and environmental *S. mutans* isolates worldwide [3,82]. Previous studies have shown that *S. mutans* exhibits limited geographic structure and high global connectivity [82–86], supporting the idea that wastewater surveillance effectively captures lineages relevant at both regional and international scales. Our findings of closely related strains persisting across nearly a year of sampling mirror those observations and suggest sustained local prevalence of a single lineage.

The *P. gingivalis* sequence from Indiana wastewater formed a moderately supported cluster with isolates from Ohio and Switzerland, indicating a close genetic relationship to North American strains and long-standing connections to European lineages. Similar phylogenetic relationships have been described for *P. gingivalis* clinical isolates, which display extensive gene flow and a globally distributed population structure [87, 88]. Detection of this keystone periodontal pathogen in municipal wastewater, although rarely reported, highlights the potential for oral bacteria to enter and persist in sewage systems. However, the lower successful amplification of *P. gingivalis* is possibly due to the significantly lower concentrations of it in wastewater over the study period. Additional studies should utilize metagenomics, nested PCR, and other methods to amplify the genome from *P. gingivalis* in wastewater samples to better understand how it compares with other previous study sequences.

Although prior studies have documented the presence of *S. mutans* and *P. gingivalis* in human feces, all reads in our study were most closely related to salivary isolates [89,90]. This finding suggests that the sequences detected in Indiana wastewater likely originated from direct shedding of oral fluids, such as saliva, rather than passage through the gastrointestinal tract. The predominance of saliva-associated strains may reflect both the higher stability or shedding of these genotypes in the sewer environment and the sensitivity of our reference database for oral rather than fecal sequences. However, since shedding studies were not performed, conclusive dominant shedding pathways cannot be determined, hence, further studies should explore importance of fecal and saliva shedding. Consequently, wastewater surveillance appears capable of capturing oral microbiome dynamics at the community level, complementing traditional fecal-based monitoring approaches.

### Limitations

A key limitation of our study is the inability to compare our wastewater data with clinical diseases that these pathogens cause (periodontal diseases and dental caries). The main reason why we cannot access this data is that there is no clinical surveillance of these pathogens in our region, which highlights how important wastewater surveillance can be to the monitoring of oral health. Additionally, while previous studies have reported the presence of these pathogens in saliva and feces, they generally do not include quantitative shedding data (e.g., genome copies per gram or per milliliter). This lack of quantitative information prevents reliable back-calculation of infected population estimates from wastewater concentrations. Furthermore,

pathogen-specific wastewater monitoring has not yet been conducted in Indiana, making it difficult to validate such estimates or assess their plausibility against local infection prevalence data. Future studies should try to incorporate clinical studies or survey data with wastewater surveillance to better connect wastewater data with the oral health of the community. To strengthen and validate the applicability of WBE for tracking dental pathogen prevalence, the recovery efficiency of the detection methodology should be assessed using spiked samples with known concentrations of target pathogens. Due to the limited availability of clinical data for dental pathogens, comparisons between observed concentrations in wastewater and expected levels based on pathogen shedding rates from sputum or feces were not done. Still, they would help to improve the estimation of infection prevalence in communities. Another comparison that future studies may consider is to include WWTPs with different dental conditions, which would provide more insight to the relationship between detection and oral health. This study did not directly assess contributions from industrial discharges or groundwater infiltration, which could influence wastewater flow independently of human population dynamics. While the sewer system is a separate sanitary system, and thus less prone to storm-related flow variability, minor contributions from infiltration or undocumented inputs cannot be entirely ruled out. Future studies incorporating flow composition analysis or additional chemical tracers may help further refine normalization approaches. Another major limitation includes the use of only molecular methods to detect the various oral pathogens in this study. While molecular methods offer fast and precise measurements for a diverse range of targets, they do not assess the viability of a specific organism. Subsequent studies should incorporate culturable or other viability assessment methods to measure the viable concentration of oral pathogens, especially when investigating persistence. Because the current study detected only genetic material, it cannot confirm an environmental transmission pathway for *S. mutans* or *P. gingivalis*, only speculate the possibility. Future work should specifically evaluate the infectivity of wastewater and environmental samples, as well as investigate whether ingestion of contaminated water could plausibly lead to infection.

## 5. Conclusion

In conclusion, this study demonstrated the detection of oral pathogen genetic material in community wastewater, indicating the potential feasibility of using wastewater-based epidemiology (WBE) for oral pathogen surveillance. The findings also highlight the need for improved public health surveillance of dental and periodontal infections, as the absence of such data limits the ability to validate wastewater observations against clinical trends. Furthermore, while this study cannot confirm an environmental transmission pathway for *S. mutans*, it underscores the importance of future research evaluating pathogen viability and possible exposure routes through environmental water. Overall, this research has the potential to provide critical insights into oral health trends, improve disease surveillance, and enhance public health strategies. By leveraging wastewater analysis alongside clinical and epidemiological data, where available, we can move toward a more proactive and non-invasive approach to oral disease monitoring, ultimately contributing to better health outcomes and reducing the global burden of dental and periodontal diseases.

## Supporting information

**S1 Table. Primers and probes for each of the targets used throughout the study.**
(DOCX)

**S1 Fig. Average wastewater flow rate per month for the local wastewater treatment plant monitored.**
(DOCX)

**S2 Fig. Comparison in bacterial concentrations between WWTP's.**
(DOCX)

**S3 Fig. Comparison in microbial concentrations between for year 1 and year 2 using Pearson's correlation 2.**
(DOCX)

## Author contributions

**Conceptualization:** Justin C. Greaves.

**Formal analysis:** Olivia N. Birch, Sang C. Par, Justin C. Greaves.

**Investigation:** Olivia N. Birch, Sang C. Par.

**Methodology:** Olivia N. Birch, Sang C. Par, Justin C. Greaves.

**Visualization:** Justin C. Greaves.

**Writing – original draft:** Olivia N. Birch, Sang C. Par, Justin C. Greaves.

**Writing – review & editing:** Olivia N. Birch, Justin C. Greaves.

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
