## [Decision Letter · Decision Letter 0]

22 Aug 2025

Dear Dr. Greaves,

Thank you for submitting your manuscript to PLOS ONE. After careful consideration, we feel that it has merit but does not fully meet PLOS ONE’s publication criteria as it currently stands. Therefore, we invite you to submit a revised version of the manuscript that addresses the points raised during the review process.

We look forward to receiving your revised manuscript.

Kind regards,

Geelsu Hwang, Ph.D.

Academic Editor

PLOS ONE

Journal Requirements:

3. We note that your Data Availability Statement is currently as follows: All relevant data are within the manuscript and in Supporting Information files.

Reviewers' comments:

Reviewer's Responses to Questions

**Comments to the Author**

1. Is the manuscript technically sound, and do the data support the conclusions?

Reviewer #1: Partly

Reviewer #2: Partly

Reviewer #3: Yes

2. Has the statistical analysis been performed appropriately and rigorously?

Reviewer #1: No

Reviewer #2: Yes

Reviewer #3: Yes

3. Have the authors made all data underlying the findings in their manuscript fully available?

Reviewer #1: No

Reviewer #2: Yes

Reviewer #3: Yes

4. Is the manuscript presented in an intelligible fashion and written in standard English?

Reviewer #1: Yes

Reviewer #2: Yes

Reviewer #3: Yes

Reviewer #1: This study demonstrated detection of the two dental pathogens in wastewater, aiming to apply WBE to the dental pathogens. Application of WBE to dental pathogens is of interest and certain novelty. However, the manuscript lacks clear focuses of the study, i.e., hypotheses to be tested, mechanisms to be understood, or concepts to be proved. Hence, the manuscript fails to provide coherent and in-depth discussion, and thus robust conclusions.

Major comments:

It seems three different studies, i.e., (i) detection of the dental pathogens in wastewater, (ii) reduction of these pathogens in a WWTP, and (iii) decay of these pathogens in wastewater, are mixed without proper coherence.

1. If the study aims to evaluate potential applicability of WBE for the dental pathogens, more in-depth study on its accuracy and validation is crucial. For example, evaluation of recovery efficiency in the detection methodology using samples with known concentration, comparison with the expected concentration which is calculated from shed quantity of the target pathogens in sputum and feces from the past clinical studies or estimated infected population from such data, comparison among different WWTPs which serve the area with different dental conditions.

2. What are the public health implications of their reduction in a WWTP? If the study wants to evaluate reduction performance, discussion on their risk is inevitable. Currently, how much of the risk is posed by WWTP effluent through exposure to the target pathogens in environmental waters receiving the WWTP effluent? And how much is the required concentration or reduction efficiency to meet the acceptable risk?

3. L.211: Calculation of normalization should be clarified in the method section. It is weird that the chart still indicated as gc/L after normalization. To normalize with the flow rate, daily total pathogen load as gc/d or gc/d/capita should be used.

4. I cannot get the purpose of the decay experiment for the main objective of this study. If it aims to evaluate the impact of sample storage, it should be discussed with methodology evaluation as commented above.

Minor comments:

1. Wastewater is suggested to include pathogens shed from sputum by several literature. I suggest to review them and include them in the introduction.

2. Figure 1 is more appropriate in the supplement. Instead, crAssphage data in Figure S1 should be shown in the main text.

Reviewer #2: This is an interesting and well-written paper that is solid in its methods, though some of the inferences are unsupported and I recommend a major revision to address the following points:

Line 33. This sentence could be interpreted as suggesting that 100 samples were taken every week. Please check and correct if that’s not the case.

Lines 41-42, 83-85 and elsewhere. Nothing in this paper or the analysis presented can speak directly to the ability of these pathogens to be transmitted by water, either drinking water or recreational exposure. This should be removed as an unsupported claim.

Introduction, lines 96-98, and elsewhere. The authors make a case that better surveillance of these pathogens could be used in targeting interventions. What sort of interventions would be appropriate? Either in the Introduction or Discussion, the authors need to provide a clear “use case” for oral pathogen monitoring by wastewater. The case should include a description of how surveillance for these pathogens is done currently and why WBE is a superior or at minimum a complementary method that can result in actionable information. Otherwise, the findings are potentially interesting but not that valuable from a public health monitoring perspective.

Reviewer #3: The author higlighted the potential of wastewater-based epidemiology to detect dental pathogens, which reflects the broader application of WBE. However, these pathogens are not strictly limited to oral diseases; some are associated with systemic conditions. This should be emphasized more precisely. Additionally, wastewater is a complex matrix, and microbial signals can originate from multiple sources, which means careful interpretation and validation are required. The decay rate experiments are informative, but many oral bacteria are atrong biofilm formers. In sewer system, attachment to pipe surfaces could protect these organisms from decay and even act as a reservoir, intermittently releasing cells. It would be more informative if the authors could comment on the biofilm-forming capabilities of these pathogens and how they influence pathogen persistence and decay rate of these pathogens.

Line 70: How strong a correlation between wastewater bacterial levels and oral disease can be established. Please explain more in details.

Line 93: What do you mean by a month-long decay experiment? Why do authors consider a month-long decay experiment? Is there any specific reason for this?

Line 108: Please provide the total catchment population if available for each WWTP.

Line 136: The authors mentioned that the flasks were covered with parafilm and left in the dark room. What about the temperature? Did the authors consider the temperature variation during the study period? If no, why? Temperature could affect the decay of the pathogens.

Line 138: Is there any reason for selecting the specific time/day for monitoring the decay rate?

Line 143: Why were all samples acidified to a PH of 3.5? Please explain it in detail, also, how it was done.

Line 151: Allprep should be replaced with AllPrep.

Line 167: Samples were tested in duplicate; how did the authors treat the one-well positive samples? Did the author consider LOD and LOQ values?

Line 211: Authors normalize pathogen concentrations using flow rate, which helps adjust for basic dilution. The authors also mentioned the stable detection of crAssphage throughout the study period. However, flow rate can vary due to other environmental factors like rainfall, industrial inputs, or groundwater infiltration, which may not reflect true changes in the human population. Did the authors evaluate how sensitive their results are to these factors, or consider whether crAssphage normalization would provide a more robust population indicator?

**Do you want your identity to be public for this peer review?** For information about this choice, including consent withdrawal, please see our Privacy Policy

Reviewer #1: No

Reviewer #2: No

Reviewer #3: No

---

## [Author Response · Author response to Decision Letter 1]

24 Sep 2025

Respones to Reviewers (Better editing is found in the attached file)

Response to Reviewer 1

The authors appreciate the reviewer for their time and consideration when reviewing the submitted manuscript.

All changes made in response to the comments have been incorporated into the revised manuscript and are highlighted for clarity.

Original reviewer comments are noted below in italics, followed by author responses

Major comments:

It seems three different studies, i.e., (i) detection of the dental pathogens in wastewater, (ii) reduction of these pathogens in a WWTP, and (iii) decay of these pathogens in wastewater, are mixed without proper coherence.

1. If the study aims to evaluate potential applicability of WBE for the dental pathogens, more in-depth study on its accuracy and validation is crucial. For example, evaluation of recovery efficiency in the detection methodology using samples with known concentration, comparison with the expected concentration which is calculated from shed quantity of the target pathogens in sputum and feces from the past clinical studies or estimated infected population from such data, comparison among different WWTPs which serve the area with different dental conditions.

Response:

The authors appreciate this insightful comment regarding the importance of assessing the applicability of WBE for dental pathogens. While we were not able to assess the shedding dynamics of target pathogens and estimate the infected population in the current area due to the gap in current research on these pathogens, we did perform sequencing analysis on the samples with the highest concentration to validate the presence of target pathogens and compare them to previous sequences. We also performed experiments on the different WWTPs in the area but could not determine differences between dental conditions in the different locations because routine monitoring for these pathogens is not done. However, our results show that the concentrations of target pathogens between the two areas are similar. We fully agree that establishing recovery efficiency in future studies is necessary and have included this in the limitations section of our manuscript.

Line 205-217: Amplicon sequencing was done for the detection of S. mutans and P. gingivalis using primers developed in this study. For S. mutans, the targeted gene was glucosyltransferase-I (gtfB) using the forward and reverse primers ATCATTACGTCTGTCCGCTATG and CATCGGCTGTCCCGTATTT, respectively. For P. gingivalis, the targeted gene was 16SrRNA and the forward and reverse primers used were CCCGTTGAAAGACGGACTAAA and CTTCAGTGTCAGTCGCAGTATG, respectively. The amplification for s. mutans resulted in a 691-nt product. The amplification of p. gingivalis produced a 606-nt product. For the qPCR, each reaction contained 10 μL of HotStarTaq® Plus Master Mix (Qiagen, Cat. No. 203645; Hilden, Germany), 2 μL of 1× HST buffer, and 5 μL of template DNA. The thermal cycling program began with a 15-minute activation at 95 °C, followed by 50 PCR cycles with denaturation at 95 °C for 5 seconds, annealing at 55 °C for 30 seconds, and extension at 72 °C for 1 minute.

Line 508-517: To strengthen and validate the applicability of WBE for tracking dental pathogen prevalence, the recovery efficiency of the detection methodology should be assessed using spiked samples with known concentrations of target pathogens. Due to the limited availability of clinical data for dental pathogens, comparisons between observed concentrations in wastewater and expected levels based on pathogen shedding rates from sputum or feces were not done. Still, they would help to improve the estimation of infection prevalence in communities. Another comparison that future studies may consider is to include WWTPs with different dental conditions, which would provide more insight into the relationship between detection and oral health.

SI Line 50-57:

Figure S2. Comparison in bacterial concentrations between WWTP’s. Top two graphs show distribution of the concentration of S.mutans (left) and P. gingivalis (right) in wastewater from DR and BP in terms of copies/L. Samples in which bacteria were not detected were removed. Bottom graphs show wastewater concentrations of S.mutans (left) and P. gingivalis (right) in DR and BP during the winter. The solid line represents the 3-sample smoothed and trimmed average.

2. What are the public health implications of their reduction in a WWTP? If the study wants to evaluate reduction performance, a discussion on their risk is inevitable. Currently, how much of the risk is posed by WWTP effluent through exposure to the target pathogens in environmental waters receiving the WWTP effluent? And how much is the required concentration or reduction efficiency to meet the acceptable risk?

Response:

The authors agree that the public health implications of these dental pathogens is important to consider and appreciate the reviewer mentioning this alongside transmission. The transmission of these dental pathogens has typically been vertical transmission through kissing or improper oral hygiene and with recent studies showing the possibility of horizontal transfer, this creates the possibility of introduction of these pathogens from wastewater into the oral cavity. The risk that these dental pathogens pose in wastewater effluent is a limitation that we have not included in our study before, but we now realize the importance of including it. These revisions have been included in the manuscript.

Line 452-461: It has been proven in past studies that the possibility of horizontal transmission of dental pathogens, like S. mutans, is possible, whether this be through kissing, shared drinks, or other forms of contact with infectious fluids (Baca et al., 2012; Momeni et al., 2016). Although the risk of transmission of dental pathogens from wastewater effluent to the oral cavity has not been fully studied, this risk can be assessed similarly to that of enteric viruses entering via the fecal oral route. When considering an acceptable or required reduction efficiency in the wastewater treatment process, future studies should examine the relationship between bacterial load and the risk of contracting an infection, as well as the infectivity of the target pathogens identified.

3. L.211: Calculation of normalization should be clarified in the method section. It is weird that the chart still indicated as gc/L after normalization. To normalize with the flow rate, daily total pathogen load as gc/d or gc/d/capita should be used.

Response:

The authors agree that more clarity should be provided in the method section on the normalization process. Concentrations were multiplied by a flow rate and population (through crAssphage) correction factor that was calculated by determining the average flow rate across the year, then dividing the monthly flow rate by the annual average. These details are placed in the text below.

Line 196-200: The flow rate and population (population normalization done through crAssphage) correction factor were calculated by determining the average flow rate or crassphage concentrations across the year then dividing the monthly values by the annual average value. Each sample concentration was then multiplied by both the flow rate and population correction factors.

4. I cannot get the purpose of the decay experiment for the main objective of this study. If it aims to evaluate the impact of sample storage, it should be discussed with methodology evaluation as commented above.

Response:

The authors appreciate this insight into the confusion and we agree that, as presented, the decay experiment does not align well with the current study. This experiment has been removed from the manuscript to improve focus and clarity.

Minor comments:

1. Wastewater is suggested to include pathogens shed from sputum by several literature. I suggest to review them and include them in the introduction.

Response:

The reviewer is correct that wastewater includes sputum, and the manuscript has not included this information, as well as appropriate literature cited.

Line 69-71: It has also been determined in previous studies that wastewater includes pathogens shed by sputum as well, which could be a strong access route for these dental pathogens (Lowry et al., 2023)

2. Figure 1 is more appropriate in the supplement. Instead, crAssphage data in Figure S1 should be shown in the main text.

Response:

The authors appreciate this insight and have now included the crAssphage data into the manuscript and the figure originally in Figure 1 is now placed in the supplementary.

Figure 1. monthly average concentrations of crAssphage in Bloomington wastewater over the two-year duration. The bars are standard deviation for four samples collected each month (n=4-5).

Response to Reviewer 2

The authors would like to thank the reviewer for their time and consideration in reviewing our paper, as well as the kind words and support.

All changes made in response to the comments have been incorporated into the revised manuscript and are highlighted for clarity.

Original reviewer comments are noted below in italics, followed by author responses

Reviewer 2 Specific Comments:

1. Line 33. This sentence could be interpreted as suggesting that 100 samples were taken every week. Please check and correct if that’s not the case.

Response:

The authors understand how the wording may be confusing and have since revised the statement for clarity.

Line 33-35: Between June 2023 and May 2025, a weekly sample of untreated wastewater was collected, resulting in a total of 100 samples collected over the timespan.

2. Lines 41-42, 83-85 and elsewhere. Nothing in this paper or the analysis presented can speak directly to the ability of these pathogens to be transmitted by water, either drinking water or recreational exposure. This should be removed as an unsupported claim.

Response:

The authors appreciate this insight and have now provided information about how previous studies have shown the possibility of horizontal transfer through individuals through kissing and improper oral hygiene. This possibility opens up the potential for it to be transmitted through water, including wastewater, but future studies should test the infectivity of these particles in wastewater as well. This information has now been included in the manuscript and appropriately cited.

Line 451-460: It has been proven in past studies that the possibility of horizontal transmission of dental pathogens, like S. mutans, is possible, whether this be through kissing, shared drinks, or other forms of contact with infectious fluids (Baca et al., 2012; Momeni et al., 2016). Although the risk of transmission of dental pathogens from wastewater effluent to the oral cavity has not been fully studied, this risk can be assessed similarly to that of enteric viruses entering via the fecal oral route. When considering an acceptable or required reduction efficiency in the wastewater treatment process, future studies should examine the relationship between bacterial load and the risk of contracting an infection, as well as the infectivity of the target pathogens identified.

3. Introduction, lines 96-98, and elsewhere. The authors make a case that better surveillance of these pathogens could be used in targeting interventions. What sort of interventions would be appropriate? Either in the Introduction or Discussion, the authors need to provide a clear “use case” for oral pathogen monitoring by wastewater. The case should include a description of how surveillance for these pathogens is done currently and why WBE is a superior or at minimum a complementary method that can result in actionable information. Otherwise, the findings are potentially interesting but not that valuable from a public health monitoring perspective.

Response:

The authors understand the reviewers’ concern and have added more information on targeting interventions and a clearer “use case” of oral pathogen monitoring in wastewater. The following text has been added to the revised manuscript.

Line 75-82: Current approaches to monitoring oral health at the population level rely heavily on dental claims data and self-reported surveys. However, these methods have notable limitations. Claims data often exclude individuals without access to dental insurance, while self-reported surveys are subject to recall and reporting biases and are typically conducted infrequently (Okunseri et al., 2023). As a result, these approaches may fail to capture timely, representative data, especially among vulnerable populations who already face barriers to dental care and experience higher burdens of oral disease.

Line 105-116: These insights could be used to inform or evaluate public health interventions such as community water fluoridation programs, school-based dental screenings, or targeted outreach in high-incidence areas(Melbye and Armfield, 2013). Importantly, integrating WBE into oral health surveillance frameworks could help address long-standing inequities by identifying underserved populations that might otherwise be missed using conventional data sources. This capability supports a more proactive and inclusive approach to public oral health planning and resource allocation. Wastewater-based epidemiology (WBE) represents a complementary and potentially more equitable surveillance tool. Because WBE captures pooled biological signals from entire communities regardless of socioeconomic or insurance status, it has the capacity to provide anonymized, near real-time insights into population-level trends in oral pathogen circulation.

Response to Reviewer 3

The authors appreciate the reviewer for their time and consideration when reviewing the submitted manuscript.

All changes made in response to the comments have been incorporated into the revised manuscript and are highlighted for clarity.

Original reviewer comments are noted below in italics, followed by author responses

Specific Comments:

1. Line 70: How strong a correlation between wastewater bacterial levels and oral disease can be established. Please explain more in details.

Response:

The aforementioned statement in the manuscript was unclear and confusing, and the authors appreciate this insight. The sentence has now been revised and improved for clarity.

Line 71-75: If a strong correlation between wastewater bacterial levels and oral disease prevalence can be established, then this approach could serve as an early warning system for emerging trends in oral health conditions, prompting timely public health interventions (Kasprzyk-Hordern et al., 2022; Mao et al., 2020).

2. Line 93: What do you mean by a month-long decay experiment? Why do authors consider a month-long decay experiment? Is there any specific reason for this?

Response:

As mentioned in the previous responses, the decay portion of the experiment has since been removed from the manuscript, as it does not align well with the other experiments.

3. Line 108: Please provide the total catchment population if available for each WWTP.

Response:

The total catchment population for each wastewater treatment plant was obtained and has now been included in the manuscript.

Line 121-123: Blucher Poole has a flow of more than 4.5 million gallons per day and caters to the northern part of Monroe County with an estimated total catchment population of 25,000 individuals

Line 128-131: Dillman Road is located on the south side of Bloomington, Indiana, and has a peak hydraulic capacity at 30 million gallons daily with an estimated total catchment population of 55,000 individuals (Bloomington, 2023).

4. Line 136: The authors mentioned that the flasks were covered with parafilm and left in the dark room. What about the temperature? Did the authors consider the temperature variation during the study period? If no, why? Temperature could affect the decay of the pathogens.

Response:

The authors understand the concern with the effect of temperature on decay and have considered it. However, the decay portion of the experiment has since been removed from the manuscript.

5. Line 138: Is there any reason for selecting the specific time/day for monitoring the decay rate?

Response

---

## [Decision Letter · Decision Letter 1]

13 Oct 2025

Dear Dr. Greaves,

Thank you for submitting your manuscript to PLOS ONE. After careful consideration, we feel that it has merit but does not fully meet PLOS ONE’s publication criteria as it currently stands. Therefore, we invite you to submit a revised version of the manuscript that addresses the points raised during the review process.

**One of the reviewers noted that the conclusions should be substantially toned down and written more carefully as commented. Please read the critique carefully and respond accordingly.** ==============================

We look forward to receiving your revised manuscript.

Kind regards,

Geelsu Hwang, Ph.D.

Academic Editor

PLOS ONE

Journal Requirements:

Reviewers' comments:

Reviewer's Responses to Questions

**Comments to the Author**

Reviewer #1: (No Response)

Reviewer #2: All comments have been addressed

2. Is the manuscript technically sound, and do the data support the conclusions?

Reviewer #1: No

Reviewer #2: Yes

3. Has the statistical analysis been performed appropriately and rigorously?

Reviewer #1: No

Reviewer #2: Yes

4. Have the authors made all data underlying the findings in their manuscript fully available?

Reviewer #1: Yes

Reviewer #2: Yes

5. Is the manuscript presented in an intelligible fashion and written in standard English?

Reviewer #1: Yes

Reviewer #2: Yes

Reviewer #1: The manuscript has been improve at some extent. However, it still fails to provide insightful findings. Moreover, main conclusions are not supported by plausible evidences. Without careful re-consideration of conclusion statements, the manuscript is not acceptable for publication.

1. The conclusion "this study demonstrated the feasibility of using WBE for oral pathogen surveillance in a community" is not proven in this study. Though detection of dental pathogens is meaningful, that does not mean its utility in WBE. In order to demonstrate its utility, its quantitative change should be verified with clinical or relevant data. The conclusion should be written more carefully by clearly distinguishing what is robust findings supported by evidence from what is merely speculated.

2. The conclusion "our study showed a potential environmental transmission pathway for S. mutans through water" is not proven in this study. Thie study only demonstrated the presence of genetic materials of the dental pathogens in WWTP effluent. We cannot prove the environmental transmission unless the followings are not proven (either in this study or in any literature): (1) the presence of not only genetic materials but also infectious viable pathogens, (2) possible infection by ingestion of water (not by mouth-to-mouth contact).

3. L.361-L.366: I wonder if these studies provide quantity of pathogen shedding from saliva and feces. If so, authors can calculate the expected population from the bacterial concentration in wastewater by assuming typical volume/mass of saliva and feces discharged daily from a person. Then, I wonder if the authors can discuss its plausibility by comparing with typical proportion of infected population from past clinical studies.

Reviewer #2: (No Response)

**Do you want your identity to be public for this peer review?** For information about this choice, including consent withdrawal, please see our Privacy Policy

Reviewer #1: No

Reviewer #2: No

---

## [Author Response · Author response to Decision Letter 2]

23 Oct 2025

Response to Reviewer 1 (Better in Attachment)

All changes made in response to the comments have been incorporated into the revised manuscript and are highlighted for clarity.

Original reviewer comments are noted below in italics, followed by author responses and changes in the manuscript.

Major Comment:

Comment:

1. The conclusion "this study demonstrated the feasibility of using WBE for oral pathogen surveillance in a community" is not proven in this study. Though detection of dental pathogens is meaningful, that does not mean its utility in WBE. In order to demonstrate its utility, its quantitative change should be verified with clinical or relevant data. The conclusion should be written more carefully by clearly distinguishing what is robust findings supported by evidence from what is merely speculated.

Response:

We thank the reviewer for this thoughtful comment and agree that our study does not establish the full feasibility or utility of wastewater-based epidemiology (WBE) for oral pathogen surveillance. As corresponding clinical data were not available, the findings cannot be directly validated against oral disease prevalence. In response, we have revised the conclusion to clarify that this study demonstrates the detection of oral pathogen genetic material in wastewater and suggests that WBE may serve as a complementary tool when used alongside clinical and epidemiological data, where available. The revised conclusion reads as follows:

Line 544-557: In conclusion, this study demonstrated the detection of oral pathogen genetic material in community wastewater, indicating the potential feasibility of using wastewater-based epidemiology (WBE) for oral pathogen surveillance. The findings also highlight the need for improved public health surveillance of dental and periodontal infections, as the absence of such data limits the ability to validate wastewater observations against clinical trends. Furthermore, while this study cannot confirm an environmental transmission pathway for S. mutans, it underscores the importance of future research evaluating pathogen viability and possible exposure routes through environmental water. Overall, this research has the potential to provide critical insights into oral health trends, improve disease surveillance, and enhance public health strategies. By leveraging wastewater analysis alongside clinical and epidemiological data, where available, we can move toward a more proactive and non-invasive approach to oral disease monitoring, ultimately contributing to better health outcomes and reducing the global burden of dental and periodontal diseases.

Comment:

2. The conclusion "our study showed a potential environmental transmission pathway for S. mutans through water" is not proven in this study. Thie study only demonstrated the presence of genetic materials of the dental pathogens in WWTP effluent. We cannot prove the environmental transmission unless the followings are not proven (either in this study or in any literature): (1) the presence of not only genetic materials but also infectious viable pathogens, (2) possible infection by ingestion of water (not by mouth-to-mouth contact).

Response:

We thank the reviewer for this insightful comment and fully agree that the current study does not demonstrate an environmental transmission pathway for S. mutans. Our data show the presence of genetic material in wastewater effluent but do not confirm pathogen viability or infectivity. We have revised the manuscript to clarify this limitation and to avoid overstating the conclusion. The text now emphasizes that further studies are required to assess the infectivity of wastewater samples and to evaluate whether ingestion of contaminated water could serve as a potential route of transmission. The revised statement can be found in the limitations section.

Line 537-541: Because the current study detected only genetic material, it cannot confirm an environmental transmission pathway for S. mutans or P. gingivalis, only speculate the possibility. Future work should specifically evaluate the infectivity of wastewater and environmental samples, as well as investigate whether ingestion of contaminated water could plausibly lead to infection.

Comment:

3. L.361-L.366: I wonder if these studies provide quantity of pathogen shedding from saliva and feces. If so, authors can calculate the expected population from the bacterial concentration in wastewater by assuming typical volume/mass of saliva and feces discharged daily from a person. Then, I wonder if the authors can discuss its plausibility by comparing with typical proportion of infected population from past clinical studies.

Response:

We appreciate this thoughtful suggestion. While the referenced studies identified the presence of the pathogen in saliva and feces, they did not provide quantitative shedding rates (e.g., genome copies per liter or per gram). Due to the lack of reported shedding concentrations, it was not possible to perform an informed back-calculation to estimate the infected population or compare it with infection prevalence from clinical data. Additionally, pathogen-specific dental monitoring has not yet been conducted in Indiana, limiting our ability to validate such estimates with local data. We have clarified this limitation in the revised manuscript

Line 508-514: Additionally, while previous studies have reported the presence of these pathogens in saliva and feces, they generally do not include quantitative shedding data (e.g., genome copies per gram or per milliliter). This lack of quantitative information prevents reliable back-calculation of infected population estimates from wastewater concentrations. Furthermore, pathogen-specific wastewater monitoring has not yet been conducted in Indiana, making it difficult to validate such estimates or assess their plausibility against local infection prevalence data.

---

## [Decision Letter · Decision Letter 2]

26 Oct 2025

Detection and quantification of key dental pathogens through wastewater monitoring

PONE-D-25-34991R2

Dear Dr. Greaves,

We’re pleased to inform you that your manuscript has been judged scientifically suitable for publication and will be formally accepted for publication once it meets all outstanding technical requirements.

Kind regards,

Geelsu Hwang, Ph.D.

Academic Editor

PLOS ONE

Additional Editor Comments (optional):

Reviewers' comments:

Reviewer's Responses to Questions

**Comments to the Author**

Reviewer #1: All comments have been addressed

2. Is the manuscript technically sound, and do the data support the conclusions?

Reviewer #1: Yes

3. Has the statistical analysis been performed appropriately and rigorously?

Reviewer #1: Yes

4. Have the authors made all data underlying the findings in their manuscript fully available?

Reviewer #1: Yes

5. Is the manuscript presented in an intelligible fashion and written in standard English?

Reviewer #1: Yes

Reviewer #1: (No Response)

**Do you want your identity to be public for this peer review?** For information about this choice, including consent withdrawal, please see our Privacy Policy

Reviewer #1: **Yes: ** Ryo Honda

---

## [Editor Report · Acceptance letter]

PONE-D-25-34991R2

PLOS ONE

Dear Dr. Greaves,

I'm pleased to inform you that your manuscript has been deemed suitable for publication in PLOS ONE. Congratulations! Your manuscript is now being handed over to our production team.

Kind regards,

on behalf of

Dr. Geelsu Hwang

Academic Editor

PLOS ONE